# Piezoelectric Based Touch Sensing for Interactive Displays—A Short Review

**DOI:** 10.3390/ma14195698

**Published:** 2021-09-30

**Authors:** Ziting Liu, Zhe Fu

**Affiliations:** Department of Engineering, University of Cambridge, Cambridge CB3 0FA, UK; liuziting_77@163.com

**Keywords:** interactive display, piezoelectric touch panel, force sensing

## Abstract

Interactive display is an important part of electronic devices. It is widely used in smartphones, laptops, and industrial equipment. To achieve 3-dimensional detection, the piezoelectric touch panel gains great popularity for its advantages of high sensitivity, low cost, and simple structure. In order to help readers understand the basic principles and the current technical status, this article introduces the work principles of the piezoelectric touch panel, widely-used piezoelectric materials and their characteristics, as well as the applications of the piezoelectric touch panel. The challenges and future trends are also discussed.

## 1. Introduction

Currently, visualization of information, which makes information easier to understand and use, plays an important role in the digital world [1]. To achieve visualization of information, the cathode ray tube (CRT), the earliest electronic display, was developed by Karl Ferdinand Braun in 1897. Since then, the electronic display has been developing quickly. Liquid crystal displays (LCD), organic electroluminescence displays (OLED), and plasma display panels (PDP) are constructed and widely used in many electronic devices, such as personal computers, smartphones, and industrial equipment [2,3,4].

With the rapid development of electronic displays, the interactive display has also made remarkable progress. Interactive display refers to the display through which the user can use his hand to display and control the data. The history of interactive display can be traced back to 1963, when the first touch panel for air traffic control was invented by E. Johnson [1]. In 1973, the first capacitive touch panel was constructed, which is very similar to the capacitive touch screen used today [5]. The first resistive touch panel appeared in the same year, which still plays a role in low-cost terminals. In 2007, Apple released the iPhone with a capacitive touch screen. The launch of the iPhone represented a milestone in the history of mobile electronic devices. Since then, the touch screen has been widely used in mobile electronic devices such as smartphones, tablets, and laptops.

One of the most important parts of interactive displays is the touch sensing part. Presently, the widely-used commercial touch panels are capacitive-based and resistive-based, both of which obtain 2-dimensional information. For the former, when the finger touches the screen, the original electromagnetic field changes and it results in the change in capacitance [6]. The capacitive touch panel is advantageous for its high detection accuracy. For the latter, when the touch events happen, the conductive layer contacts the other and hence the current flows to it. By sensing the current, the touch location can be interpreted [7]. Contrary to the capacitive touch panel, the touch of the non-conductive subject is also supported. However, both of them can only obtain 2-dimension information. To add the function of force sensing, an extra detection layer or sensor should be added, and therefore the structure and circuity will become bulky, costly, and complex, which is not desirable.

To obtain 3-dimensional information in a low-cost, piezoelectric touch panel is a suitable approach. In the piezoelectric-based touch panel, the piezoelectric material can be used as the insulating layer of the capacitive touch panel [8,9]. As a result, it can obtain force information in a much simpler architecture. In addition, the piezoelectric touch panel can provide benefits such as passive detection and high sensitivity [10]. However, the piezoelectric touch panel was rarely used in commercial products, mainly due to the temperature and frequency dependency and the unstable responsivity introduced by the different contact area and touch direction.

After years of development, with its excellent performance, the piezoelectric touch panel has attracted extensive attention in industry and academia. Many researchers devote themselves to it, and their achievements also attract more new researchers to work on it. However, the piezoelectric touch panel covers many fields, including materials science, computer science, and others, and there is no systematic review in the published literature, making it difficult for the beginners interested in this field to begin their work. In this context, this article is presented to provide a comprehensive review on piezoelectric-based sensing for interactive display.

This review is organized as follows: Section 2 briefly reviews the principle of piezoelectricity and the widely used piezoelectric materials; Section 3 introduces the commonly used architecture and readout circuity of the piezoelectric touch panel; The application of the piezoelectric touch panel as well as the drawbacks and solutions are reviewed in Section 4; Section 5 introduces the emerging applications based on the piezoelectric touch panel; Section 6 reviews the challenges that piezoelectric touch panel is facing. In Section 7, the conclusion and perspective are discussed.

## 2. Principle and Broadly Used Materials

### 2.1. Principle

The piezoelectric effect demonstrates the phenomenon that when a force is applied to non-centrosymmetric materials [11], the centers of the positive and negative charge do not coincide, thus resulting in polarization and the induced charge. In contrast, this phenomenon does not occur in centrosymmetric materials, as the polarization always remains intact. In non-centrosymmetric materials, the phenomenon inverse piezoelectric effect also exists, which indicates that when an external electric field is applied to the materials, the materials will deform [12].

Due to the various directions of the applied force and the polarization, axes should be built to describe the parameters. Normally the model is built as shown in Figure 1.

The relationship between the polarization and the applied stress can be described as below [13]:(1)$$P_{i}{=d}_{ij}\sigma_{j}\;\mathrm{with}\;i=1,\;2,\;3\;\mathrm{and}\;j=1,\;2,\;\ldots ,\;6$$
where $P_{i}$ indicates the polarization in the i-direction, ${\;d}_{ij}$ is the piezoelectric coefficient, and $\sigma_{j}$ is the applied stress in the j-direction. For example, the parameter ${\;d}_{33}$ is used when the polarization and the applied stress are both in the “3-direction”; ${\;d}_{33}$ is essential to quantify the piezoelectric performance, which can be measured by using the Berlincourt method [14].

### 2.2. Broadly Used Materials

#### 2.2.1. Piezoceramic

Piezoceramic is now the most popular piezoelectric material due to its high efficiency and low cost, including lead zirconate titanate (PZT), barium titanate (BaTiO_3_), among others. As the Young’s modulus in piezoceramics is higher than in other materials, it has stronger rigidity, which makes it more brittle and less flexible for design [15]. Lead-based piezoceramics like PZT have a high piezoelectric coefficient and low dielectric loss. Therefore, it is widely used in energy harvesting devices, which convert mechanical energy into electrical energy. However, due to the lead it contains, the extraordinary toxicity limits its use [16]. Despite the lower transduction efficiency, lead-free materials like BaTiO_3_, which is non-toxic, is gradually replacing PZT [17].

#### 2.2.2. Single Crystal Materials

Quartz is currently the most widely used piezoelectric crystal. The electric resistivity of quartz is excellent. It also has an extremely high mechanical quality factor Q_M_, as well as temperature stability, making it commonly used in resonators. The main disadvantages include lower piezoelectric coefficients and higher costs. Piezoelectric crystals of lithium niobate (LiNbO_3_) and lithium tantalate (LiTaO_3_) are also excellent materials for piezoelectric sensors. Specifically, the lower acoustic losses making them outstanding materials that are widely used in surface acoustic wave (SAW) devices [18].

However, the phase transition temperatures of the single crystals place restrictions on the temperature range of their applications, making it difficult for them to be applied in extreme conditions [19].

#### 2.2.3. Piezoelectric Polymers

One of the challenges piezoceramic and single crystal materials face is the high Young’s modulus and the resulting brittleness. In 1969, the discovery of piezoelectricity in polymers represented a milestone in soft piezoelectric materials [20]. Due to the flexibility in the molecular chains, piezoelectric polymers, for instance, polyvinylidene fluoride (PVDF) and polyvinyl chloride (PVC), are much more flexible. Therefore, they can endure the high strain and can be applied to a large area. Additionally, piezopolymers are cost-effective and easier to process when compared with piezoelectric ceramics. However, the Curie temperature of piezoelectric polymers is much lower than those of piezoceramic and single crystal materials [21], which is a great limitation to their applications.

Piezoelectric polymers are widely used in many fields. They are used for energy harvesting applications [22,23], flexible touch sensing [24], biological tissue engineering, and so on.

#### 2.2.4. Ceramic-Polymer Composite

As discussed previously, piezoelectric ceramics possess high piezoelectric coefficients; however, the poor flexibility of ceramics limit their application. In comparison, piezoelectric polymers are much more flexible but have smaller piezoelectric coefficients. To meet the requirement of both good flexibility and high performance, researchers have proposed ceramic-polymer composites, in which the ceramic phase is dispersed in a polymer matrix [25]. This kind of material shows excellent properties of good flexibility, high piezoelectric coefficients, and low density. Moreover, in [26], an interfacial chelation mechanism between PZT and a chelating polymer is reported, which can improve the dielectric constant and piezoelectric coefficient of ceramic-polymer composites significantly.

Ceramic-polymer composites are widely used in many fields. With their flexibility and good piezoelectric properties, they are applied to piezoelectric nanogenerators with high output performance [27]. In addition, as the polymer phase lowers the density and dielectric constant, ceramic-polymer composites are easier to realize acoustic impedance matching compared to ceramics, and therefore they are widely used in hydrophones, medical examination probes, and sonar [28].

#### 2.2.5. Piezoelectric Fabric

Piezoelectric fabric is popular in the field of smart textile and wearable applications due to its flexibility. The major advantage of piezoelectric fibers is that they can be directly made into large fabrics, which is difficult for other materials [29]. In addition, compared to piezoelectric polymers, the fabrication process of piezoelectric fabric is simpler. To fabricate piezoelectric fibers, the electrospinning method is most commonly used. Electrospinning is widely used to make polymer fibers, as it is possible to form various fibers of different properties, shapes, and diameters. However, this method is time- and cost-consuming, and therefore it is not suitable for mass production [30].

Piezoelectric fibers can be widely used in wearable applications [31]. Considering the rigidity and wearability of piezoelectric fibers, it is the desirable material applied to smart textiles, particularly for wearable energy harvesting [31].

The properties of the commonly used piezoelectric materials mentioned above are shown in Table 1. As for new piezoelectric materials that may be used in future interactive displays, the flexible piezoelectric materials have become the focus of research, as one of the important trends of interactive displays is flexible display and wearable devices. In [32], new all-silicone composite materials are reported. Cyanopropyl-modified polysiloxanes and chloro-modified polysiloxane are crosslinked with non-polar PDMS to produce the materials. Young’s modulus of the highly flexible material is 0.12–0.5 MPa, and the average piezoelectric coefficient is 24 pm·V^−1^, close to that of PVDF. In addition, ceramic-polymer composites are receiving great attention due to their flexibility and good piezoelectric properties. In [33], highly flexible piezoelectric composites with poly dimethyl siloxane (PDMS) using herbal zinc oxide (h-ZnO) as filler are reported. The Young’s modulus of the material is 16 MPa and the piezoelectric coefficient is 29.76 pm/V (h-ZnO 30 wt.%), showing both great flexibility and good piezoelectric properties. In [34], PZT-PDMS composites prepared by solution casting were developed. The piezoelectric coefficient of the material achieves 25 pC/N, and Young’s modulus is 4.85 MPa. Hopefully, these highly flexible piezoelectric materials can be applied to flexible displays in the future.

### 2.3. The Growth of Piezoelectric Materials

Among the piezoelectric materials, the piezoelectric thin film is the most widely used form, which has the advantages of small volume, low weight, high working frequency, and a simple manufacture process for the multi-layer structure. Piezoelectric thin films are widely applied to piezoelectric microelectronic systems and optoelectronic technology fields. The most widely used piezoelectric thin film is of PZT piezoelectric ceramics, which, however, is fragile, highly toxic, and can cause damage to humans. To overcome this problem, lead-free piezoelectric thin films like BaTiO_3_, KxNa1−xNbO3(KNN), and Bi0.5Na0.5TiO3(BNT) are proposed. In addition, piezoelectric thin films based on polymers like PVDF are also receiving great attention due to their good flexibility.

To fabricate piezoelectric thin films, the magnetron sputtering method, sol-gel method, vapor deposition method, and solvothermal method are widely used [38]. Magnetron sputtering refers to the process of bombarding the target with high-speed moving inert particles to deposit atoms on the substrate to form a thin film. This method can form thin films with a large area and good uniformity. However, its high manufacturing cost is not desirable. The sol-gel method dissolves a metal salt in a common solvent and forms a uniform precursor solution (SOL) after hydrolysis and polymerization, then coats the SOL on a substrate surface. After drying and repeated coating, the film is finally processed by annealing. This method is simpler, but the thin film easily cracks under high-temperature sintering. The vapor deposition method applies the compound or elemental gas to the substrate, and the film is formed by gas-phase action or chemical reaction on the substrate surface. This method can obtain high purity and dense films; however, it is inconvenient because different cavities are required to form different forms. The solvothermal process obtains thin films by chemical reaction with organic matter or a non-aqueous solvent as the solvent is under high temperature and high pressure. It has the advantages of a simple process and low temperature. However, as the process is under high pressure, it is dangerous and needs sophisticated equipment. The typical growth methods for the preparation of BaTiO_3_ thin films are shown in Table 2.

## 3. Piezoelectric-Based Touch Sensing Architecture

### 3.1. Typical Architecture

The most widely used architecture in the piezoelectric touch panel is the bending acoustic wave-based architecture. There are three or four piezoelectric receivers arranged on the edge of the panel to receive the mechanical waves. According to the transmission speed and the timing of the mechanical waves, the touch location can be evaluated. However, it only provides 2-dimension information [42].

To obtain 3-dimension information, including the x-y position and the force amplitude, the most widely used architecture is the sandwich architecture [42]. In this architecture, two electrodes are attached to both sides of the piezoelectric material. This typical architecture is shown in Figure 2.

There are four proposed stack-ups to assemble touch panels as shown in Figure 3. As the electrode layers are much thinner when compared to the other layers, they are not described in the figure. Figure 3a–c gives the designs that are similar to the capacitive touch panels [7]. The insulating layers of the capacitive touch panels are changed to the piezoelectric ones to fabricate the piezoelectric touch panels. These designs are advantageous for the high sensitivity in detecting the force, but they have low accuracy when detecting the x-y position [9,44]. To solve this problem, another design is put forward. Figure 3d shows a design that separates the force layers and the capacitive layers, in which the 2D position and the force amplitude can be both obtained at high accuracy [45]. The former is detected by the capacitive layers whereas the latter can be interpreted by the piezoelectric layers.

### 3.2. Typical Readout Circuity

The 3-dimensional touch panel consists of capacitive sensing and piezoelectric sensing. The piezoelectric sensor is equivalent to a charge generator, a resistance $R_{\mathrm{PF}}$ and a capacitance $C_{\mathrm{PF}}$. The generated charge is determined by the applied force and the piezoelectric coefficient $d_{33}$, while the resistance and the capacitance are determined by the dielectric constant, resistivity, and geometry. The equivalent circuity of the piezoelectric sensor is shown in Figure 4.

The readout circuity of the piezoelectric sensor is shown in Figure 5. The charge amplifier is used to convert the charge signal into the voltage signal. For piezoelectric sensing, charge generates when the touch events occur; For capacitive sensing, when the touch events occur, the capacitance will increase. Under the same voltage, the generated charge will change.

The block diagram of piezoelectric touch panel system is shown in Figure 6. As the frequency of the piezoelectric signal is 0–10 kHz [42] while the frequency of the capacitive signal is around 100 kHz, they can be separated smoothly by using a low pass filter and a bandpass filter. The ADC is then used to convert the analog signal to a digital signal, which is then calculated to interpret the force amplitude and the touch location by the signal processor.

## 4. Conventional Applications and Discussions

### 4.1. Conventional Piezoelectric-Based Touch Panels

The piezoelectric-based touch panel, developed rapidly in recent years, can be divided into 2-dimensional touch panels and 3-dimensional touch panels. As mentioned above, for the 2-dimensional touch panel, the piezoelectric receivers sense the bending acoustic wave and interpret the touch location accordingly. However, due to the high dispersion characteristics of the waves, surface bending waves that have different frequencies propagate at different speeds, which means that using a fixed speed for calculations will decrease the detection accuracy. To address this issue, the algorithms of acoustic pulse recognition (APR) and dispersive signal technology (DST) were developed. For the former, the received signals are matched with the models in the database to interpret the touch location. It has been commercialized in the products of Elo TouchSystems. For the latter, a specific program is utilized to calibrate the errors introduced by the frequency. In 2006, 3M first launched the commercial product based on DST technology.

For the 3-dimensional touch panel, the force amplitude and the touch location are interpreted by the combination of piezoelectric-based techniques and capacitive-based techniques. As the piezoelectric sensing layer can act as the insulating layer of the capacitive touch panel, the structure is simpler compared to capacitive and resistive 3-dimensional touch panels. In addition, it has the advantages of passive detection and high sensitivity. In [13], a 3-dimensional piezoelectric touch panel was developed, and 0.2 pF of the minimum changed capacitance value and 92 mV/N of the force-voltage sensitivity were achieved, revealing the great potential of piezoelectric touch panels. In 2016, a 3-dimensional touch panel based on piezoelectricity was issued by Cambridge Touch Technologies Ltd. (CTT, Cambridge, UK). However, piezoelectric-based 3-dimensional touch panels have rarely been applied to commercial products, mainly due to the unstable responsivity introduced by the contact area and touch speed, as well as the interference from the external environment.

### 4.2. Drawbacks and State of the Art Solutions

#### 4.2.1. Contact Area and Touch Direction

When the same force is applied to different contact areas on the piezoelectric-based touch panel, the stress-induced signals will also be different, leading to unstable responsivity. As the diameter of fingertips can vary from 7 mm to 15 mm, the responsivity will greatly change due to the different size of fingertips.

Furthermore, the different orientations of the same applied force will also result in different responsivities, as the force closer to the perpendicular orientation will obtain more stress in the z-direction. In the study of [42], it was shown that the responsivity increases by 51% when the touch orientation increases from 30° to 90° as shown in Figure 7.

One effective method to solve this problem is to estimate the contact area and touch orientation and then calibrate the force accordingly. In [46], the contact area and touch angle are measured by utilizing the capacitance distribution information, improving the stability of force voltage responsivity to 85%. In [45], an artificial neural network is applied to classify the touch angles, achieving a force detection accuracy of 90%. Nevertheless, as the data in the study are collected in the lab, the accuracy of the technique may decrease in practical use because environmental noise will pollute the data. To address this issue, in [47], a noise robustness technique is presented. Different levels of noise are added to the training data to simulate the noisy environment. As a result, the average mean error (MAE) is 7.8 degrees, improving the performance of the piezoelectric touch panel used in a noisy environment.

#### 4.2.2. Touch Speed

The piezoelectric coefficient $d_{33}$ is frequency dependent. Before reaching the resonant frequency, the value of $d_{33}$ changes little as the frequency increases, and therefore can be regarded as a constant. However, when approaching the resonant frequency, $d_{33}$ changes significantly, leading to a sharp increase in responsivity [48]. As the frequency of the force touch signal ranges from DC to kHz, the change of $d_{33}$ cannot be ignored.

One method to solve this problem is to calibrate $d_{33}$ by obtaining the frequency information. In [49], the obtained signal is divided into several intrinsic mode functions (IMF) by using the empirical mode decomposition (EMD), and then the normalized Hilbert transform is used to interpret the frequency information. Finally, the $d_{33}$ coefficient is calibrated at every moment accordingly. The detection accuracy is improved by 2.5% using this method.

How the contact area, touch direction and touch speed are leading to the unstable responsivity is shown in Figure 8.

#### 4.2.3. Propagated Stress

As shown in Figure 9, when a force is applied to the touch panel, the stress will propagate to the adjacent locations and therefore reduce the detection accuracy. The responsivity induced by the propagated stress cannot be neglected, as it can even be higher than that at the actual touch locations [9]. One method to solve this problem is to eliminate the propagated stress by using capacitive signals. In [50], the output with capacitive touch is augmented, as the propagated stress only causes touch-induced charge without a change in capacitance. In this way, the propagated stress can be eliminated successfully.

#### 4.2.4. Preload Effect

The static force applied to the piezoelectric material will result in the deformation of the molecular structure, leading to the change of the force-to-voltage conversion ability. Therefore, when a prestress is applied to the piezoelectric touch panels, the piezoelectric coefficient will change, affecting the detection accuracy in the subsequent touch. In [51], the responsivity with different preload is measured and the results are shown in Figure 10. One effective method to solve this problem is to calibrate the piezoelectric coefficient under different preloads. The essential step of this method is to measure the prestress. In [51], the prestress is obtained by using the change of the resonant characteristics of the piezoelectric touch panels. The accuracy is boosted by 15.17% after applying the algorithm.

#### 4.2.5. Boundary Condition

Due to the complexity of the mechanical behavior and the boundary conditions of the touch panel, the responsivity can be different when the same force is applied to different locations, leading to stress non-uniformity and low detection accuracy. In [8], the mechanical response of the different locations was investigated. As shown in Figure 11a, Location 1 is near the edge of the touch panel, while Location 5 is at the center. As the results show in Figure 11b,c, the different boundary conditions result in different mechanical responses, which may reduce the detection accuracy.

To solve the non-uniformity issue, in [52], a set of artificial neural networks is applied to estimate the force location and classify the force amplitude. As a result, the detection accuracy achieves 94.2%. As the curvature is also essential to the responsivity, in [53], the capacitive information is used to measure the curvature radius and then calibrate the force responsivity in real-time.

The drawbacks of the piezoelectric touch panels and the performance of the solutions mentioned above are shown in Table 3.

## 5. Emerging Applications

### 5.1. Authentication

With the quick development of information technology and wide use of smartphones, user authentication is becoming a hot topic, of great importance for maintaining information security [54]. As force information is greatly appropriate for user authentication, the piezoelectric touch panel can be applied to authentication. In [55], the piezoelectric touch panel was utilized to collect user touch data, and then the user touch times and force features are extracted from the data by machine learning for authentication. As a result, the equal error rate (EER) of 0.720% is achieved, indicating the great potential for the piezoelectric touch panel to be used in authentication. However, multiple users should be treated as legal users in some applications. To address this issue, in [56], a piezoelectric-based technique for multi-user authentication is presented, in which the touch features are extracted from the information obtained from the piezoelectric touch panel, and the classification is done by a support vector machine (SVM). Finally, the classification accuracy reaches 97%, revealing the feasibility for multi-user authentication.

### 5.2. Underwater Application

There is also a need for touch panel applications in extreme conditions, such as underwater applications. The properties of a capacitive touch panel will be greatly affected when exposed to water, as the ions contained in the water will affect the electromagnetic field and reduce the detection accuracy as a result. In contrast, the detection accuracy of the piezoelectric touch panel will not be affected by a small amount of water, as it is sensitive to the mechanical displacement instead of the capacitance change. In [44], a piezoelectric touch panel was successfully developed that can work without interference underwater. The promising results reveal the potential for the piezoelectric touch panel to be applied to underwater applications.

### 5.3. Mood Detection

Mood detection is receiving extensive interest in the field of human-machine interface. Among the related studies, the keystroke-based technique that extracts user’s touch behaviors, such as touch time and force amplitude, to detect emotion is advantageous for its low cost [57]. However, using touch sensing products that have low accuracy will result in long-term text input, and therefore limits its use [58]. Due to the high sensitivity of the piezoelectric touch panel, it can achieve high accuracy without long text input and is a suitable approach for detecting emotion. In [59], the piezoelectric touch panel was utilized for extracting keystroke behaviors, including touch time, pressure value, and the direction of pressure. The emotion classification was completed by the random forest classifier, and an accuracy of 78.31% was achieved. This result is promising and proves the potential for the piezoelectric touch panel to be used in mood detection.

### 5.4. Multi-Touch

Most current force touch panels interpret the touch position and amplitude of the single force. However, multi-touch, which means more than one finger touches the panel at the same time, is required in many applications [60]. Due to the unstable responsivity of the piezoelectric touch panel, the physical model of multi-touch is difficult to establish, and therefore it is difficult to apply formulas to interpret multi-touch events. In [61], this problem was solved by using machine learning methods. After pre-processing the collected data to get the voltage peaks of each channel, three machine learning methods were applied to classify the touch position and the force levels. As a result, the accuracy of double-touch force level detection achieved 88.2%, and the accuracy of location detection achieved 92.3%, proving the feasibility of the piezoelectric touch panel to detect multi-touch events.

### 5.5. Robotic Skin

The concept of the flexible display was first proposed in 1974. After that, many commercial products of flexible display have been launched, and the function of touch interactivity was also added to the flexible display. At present, the flexible piezoelectric touch panel has been successfully applied. In [62], a flexible and transparent piezoelectric touch sensor based on Zinc Oxide (ZnO) nanowires was presented that had good piezoelectric properties and bendability. Even after 2000 times of bending, the output voltage of the presented sensor remained stable, showing the feasibility for the piezoelectric touch panel to be applied to flexible and wearable devices.

The development of the flexible piezoelectric touch panel brings piezoelectric-based electronic skin feasibility. Electronic skin is an essential part of humanoids, as it contributes to protecting the inner system and collecting signals [63]. As the electronic skin should be applied to large area and be able to detect force, the flexible piezoelectric-based sensing technique is a suitable approach. As the piezoelectric-based electronic skin also measures the touch position and force amplitude, it can use a similar structure as the piezoelectric touch panel. In [64], a large-area flexible artificial skin based on arrays of piezoelectric polymer transducers was developed. The tactile sensing system consists of the arrays of PVDF piezoelectric sensors, which are fixed on a flexible substrate. The promising results show the feasibility of the piezoelectric-based artificial skin, with the advantages of bendability, scalability, and low fabrication costs. Furthermore, due to the coupling ability of the piezoelectric technique with the capacitive technique, electronic skin that supports multi-functions can be developed. In [65], a humanoid skin with the combination of the piezoelectric and capacitive effects successfully detected the position, force, humidity, and proximity. The force and capacitive sensitivities achieved 0.05 N and 7 fF, and the humidity responsivity achieved 0.22%/RH%.

### 5.6. Gait Analysis

Gait analysis has received great attention recently, as it can monitor the health problems of the elderly (like Parkinson’s disease and cerebral palsy) as well as the walking status of the patients with walking problems. Among the sensors for gait analysis, the piezoelectric sensor is advantageous for its passive detection and low power consumption [66]. As the piezoelectric sensor used for gait analysis also measures the touch position and force amplitude, it can use a similar structure as the piezoelectric touch panel. In [67], a pair of slippers with flexible PVDF sensors was developed. A flexible sensor based on PVDF was installed in the slipper sole area under the inner arch of the foot. When the subject walked, normal pressure is generated and therefore the piezoelectric sensor under the inner arch of the foot generated output voltage. Eight gait parameters, including the time parameters and the number of steps, were then extracted from the sensing signals by detecting the heel strike and toe-off times. The promising results show the good consistency between the gait parameters from the program and the parameters calculated using the raw data. However, it is difficult to mass produce. In [68], the lamination technique was used to address this issue. It can integrate electrode and piezoelectric thin film without affecting the flexibility or conductivity of the film. Finally, it can successfully detect the normal forces at 36 points, and the responsivity achieves 693.1 mV/N while the sensitivity achieves 0.056 N.

The emerging applications based on the piezoelectric touch panels mentioned above are shown in Table 4.

## 6. Challenges

Except for the drawbacks of the piezoelectric touch panels mentioned above, there still remain challenges for their applications, including the changes of the external conditions, the issues introduced by multi-touch, and so on. These challenges are discussed in this section.

### 6.1. Pyroelectric Effect Interference

Pyroelectricity is the phenomenon that when there is a temperature change in the non-centrosymmetric materials, the polarization alters and generates a charge on the surface [69]. It should be noted that some piezoelectric materials also have pyroelectric properties, and piezoelectricity and pyroelectricity are closely related as they are both relevant to polarization. Therefore, when a touch event happens on the piezoelectric touch panel, both the pressure and the temperature change will alter the output voltage: the former causes a piezoelectric effect and the latter causes a pyroelectric effect. It is hard to separate these signals as they are generated at the same time [70]. As a result, the pyroelectric effect will influence the detection accuracy of the piezoelectric touch panel, especially when there is a big difference in temperature between the finger and the touch panel. Although adding a thermal-shielding layer on the touch panel can reduce the pyroelectric effect [71], the reduction of the optical transparency and the increasing cost it brings are undesirable. Therefore, it still remains a challenge for the piezoelectric touch panel to remove the pyroelectric effect.

### 6.2. Triboelectric Effect Interference

Triboelectricity is the phenomenon of when two different materials are brought into contact, they will obtain or lose electrons and be electrically charged, due to the different binding abilities of the materials to extranuclear electrons. There are two fundamental modes of triboelectricity: vertical contact-separate mode and lateral sliding mode [72]. In contact-separate mode, when an external force is applied to one of the surfaces, the two surfaces come into contact and generate charge, due to surface charge transfer. In the lateral sliding mode, the relative sliding of the two surfaces will generate triboelectric charges. When there is a touch event on the piezoelectric touch panel, both the contact-separation process and the sliding motion will happen simultaneously, generating the triboelectric charges. As a result, triboelectricity will affect the output voltage and therefore reduce the force detection accuracy. Therefore, in order to achieve higher detection accuracy, eliminating the triboelectric influence is a significant challenge for the piezoelectric touch panel.

### 6.3. Sensitivity to Temperature Change

After reaching a particular temperature called the Curie point, the piezoelectricity of the piezoelectric materials will decrease, due to the phase transition from a ferroelectric to a paraelectric and the resulting lost polarization [73].

Nevertheless, even under the Curie point, piezoelectric materials show great temperature dependency. Due to the increasing ionic movement under high temperature, the piezoelectric constant significantly rises with temperature. In [73], the piezoelectric response of the poly(vinylidene fluoride70-trifluorethyline3) was investigated. The results showed that piezoelectric constant of the polymer increased by 150% when the temperature increased from 25 °C to 80 °C, demonstrating a strong temperature dependency. In [74], the variation of the piezoelectric constant with temperature was measured. The result not only showed the temperature dependence of the piezoelectric properties, but also indicated the non-linearity of the piezoelectric coefficient under high temperature. As a result, the variation of the ambient temperature has a significant effect on the piezoelectric response, influencing the detection accuracy greatly. How to eliminate the interference introduced by the temperature change still remains a challenge.

### 6.4. Multi Touch Interference

The multi-touch panel is now receiving increasing attention. However, when multiple touches are applied to the piezoelectric touch panel simultaneously, more complicated issues can arise. As mentioned above, due to the boundary conditions, the response of the propagated stress can be even higher than the response of the real touch location. When multi-touch is applied to the touch panel, the detection accuracy can be affected more significantly. In [9], it was demonstrated that when the touch events occur at adjacent positions on the touch panel simultaneously, the voltage shift reached 27 mV, decreasing the detection accuracy. As shown in Figure 12, the adjacent force disturbs the response. Additionally, the preload induced by the first applied force can affect the detection of the following force significantly, due to the change of the piezoelectric coefficient. As a result, it is still difficult for the piezoelectric touch panel to achieve multi-touch.

## 7. Conclusions and Perspective

At present, most capacitive and resistive touch panels widely used in commercial products obtain 2-dimensional information. To obtain the force amplitude to achieve more functionality, the piezoelectric touch panel is a suitable approach. Studies show that, as the piezoelectric material can act as the insulating layer in the capacitive touch panel, it has a simpler structure; in addition, due to the natural characteristics of the piezoelectric material, it has the advantage of passive detection and therefore has a lower power consumption. An increasing number of studies of the piezoelectric touch panel show its advantages and great potential.

However, the piezoelectric touch panel is still not widely used in commercial products, mainly due to the unstable force-voltage responsivity. Therefore, an important direction of future research is to improve the stability of the responsivity. The most important step is to obtain the information of the interference factors, like the touch direction and the touch area. One effective approach is to utilize multidimensional information, such as the capacitance information and the resonant information to calibrate the responsivity. Another method is to use machine learning. The quick development of machine learning makes it possible to calculate the interference factors and calibrate the responsivity accordingly.

In addition, as the problems of global warming and environmental pollution are becoming more and more serious, using renewable energy instead of traditional fossil energy to maintain sustainable development is important for the technology. The preferred approach is to use the piezoelectric nanogenerators (PENG) as the power resources to achieve self-powered systems. Recently, Sumera et al. reported Br doped 2D ZnO PENG, which is used as an active self-powered pressure sensor, for the measurement of a wide range of pressure [75]. In the future, it is expected that PENG will be integrated into the piezoelectric interactive displays to achieve a self-powered system.

Furthermore, if the piezoelectric touch panel is widely used in the future, the 3-dimensional information it measures will make it possible to be used in many emerging applications. One example is the piezoelectric touch panel used for authentication. As the process of the authentication based on keystroke is continuous, the security level is higher; in addition, no extra component is needed, so it has a simpler structure and lower cost. The piezoelectric panel can also be used for mood detection, and it has the advantages of being cost-effective and non-intrusive. In the foreseeable future, the piezoelectric touch panel can be widely used in the applications in which force information is needed.

## Figures and Tables

**Figure 1 materials-14-05698-f001:**
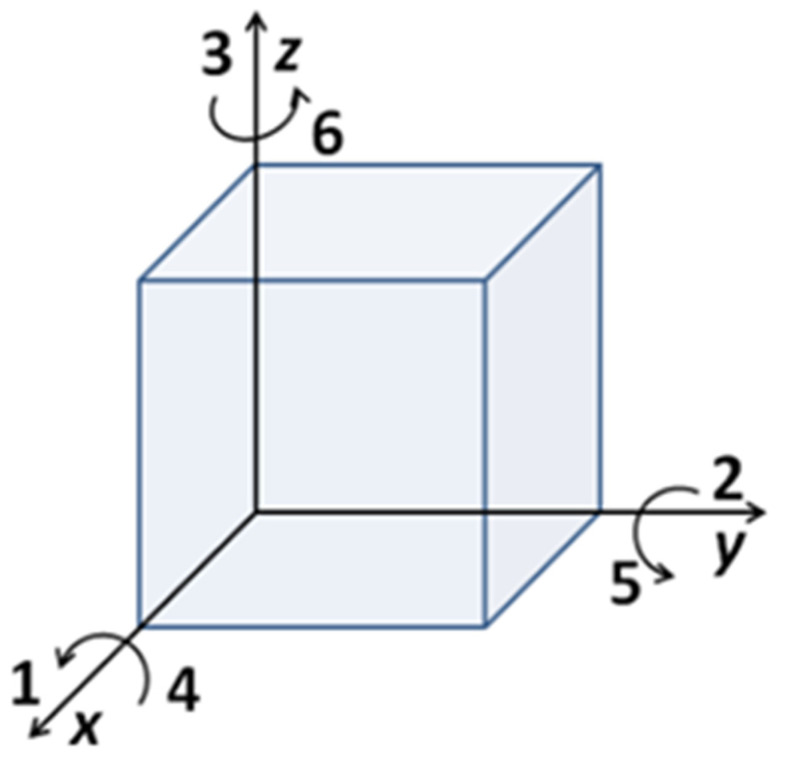
Directions of the polarization and the applied stress.

**Figure 2 materials-14-05698-f002:**
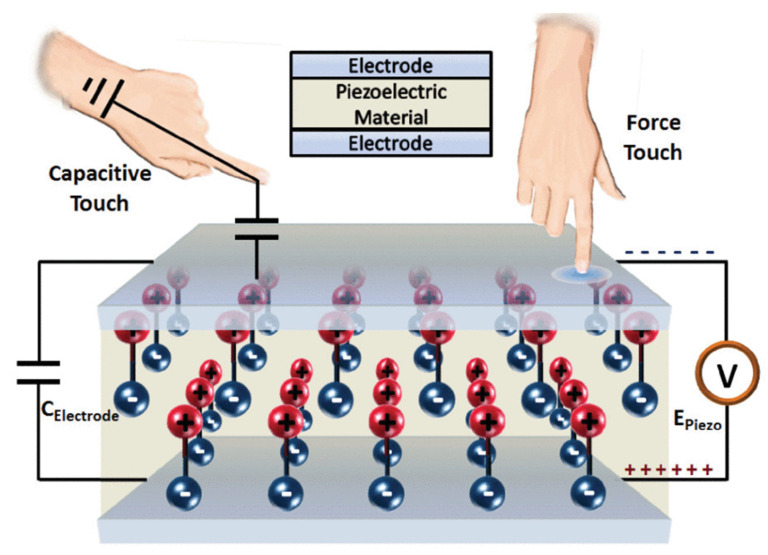
The sandwich architecture of the piezoelectric touch panel [43].

**Figure 3 materials-14-05698-f003:**
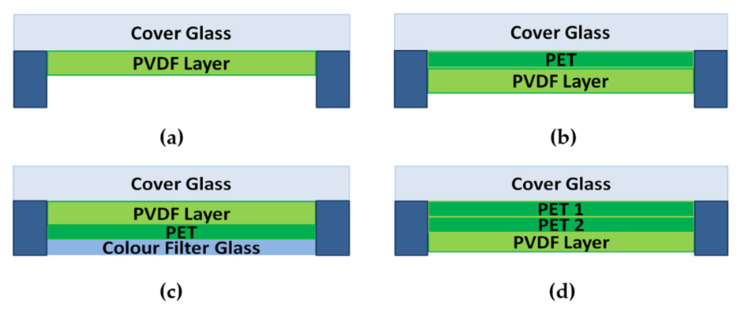
Four typical stack-ups of the piezoelectric touch panel: (**a**–**d**), the electrodes are on and underneath the piezoelectric film layer [42].

**Figure 4 materials-14-05698-f004:**
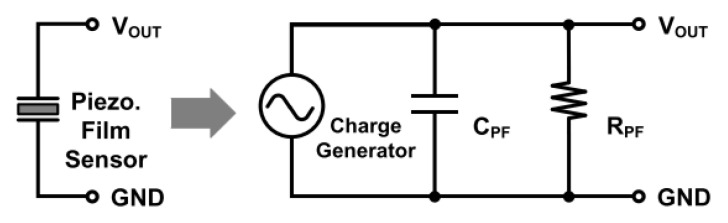
The equivalent circuity of the piezoelectric sensor [13].

**Figure 5 materials-14-05698-f005:**
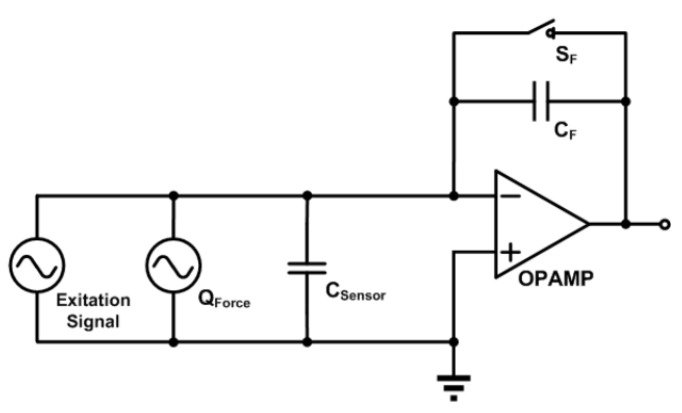
Readout circuity [13].

**Figure 6 materials-14-05698-f006:**
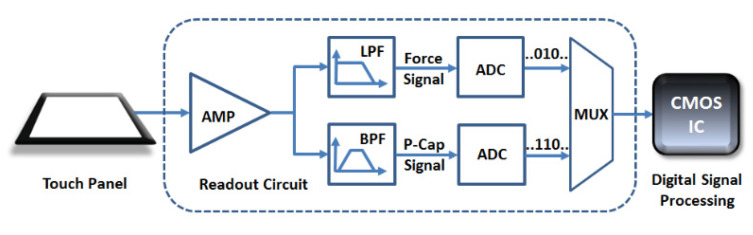
Block diagram of piezoelectric touch panel system [13].

**Figure 7 materials-14-05698-f007:**
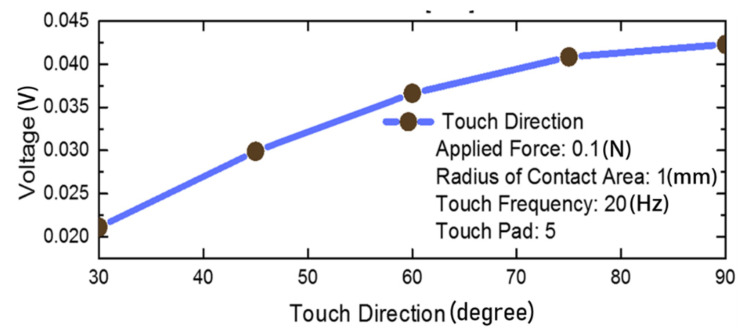
The voltage response with the variation of the touch direction [42].

**Figure 8 materials-14-05698-f008:**
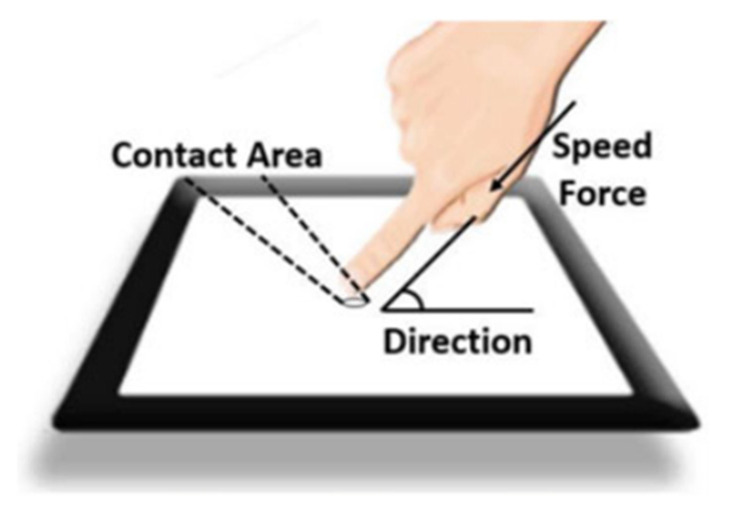
Three factors leading to unstable force-voltage responsivity [42].

**Figure 9 materials-14-05698-f009:**
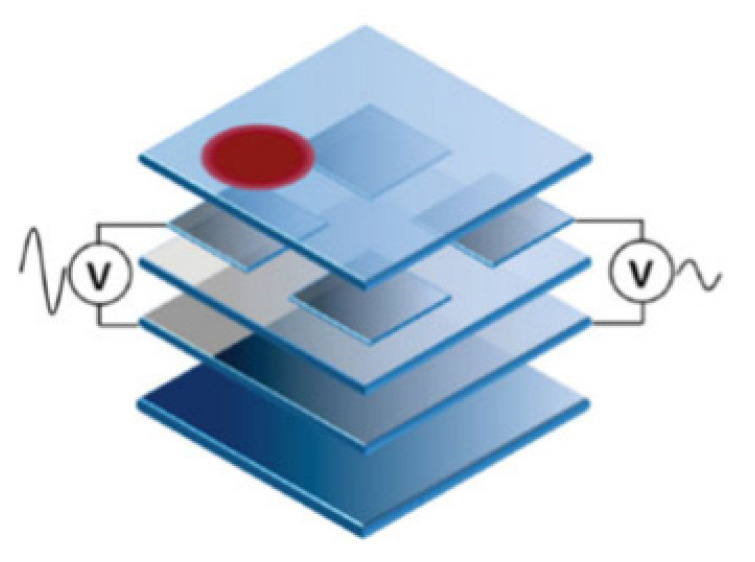
The propagated stress is related to the detection accuracy [42].

**Figure 10 materials-14-05698-f010:**
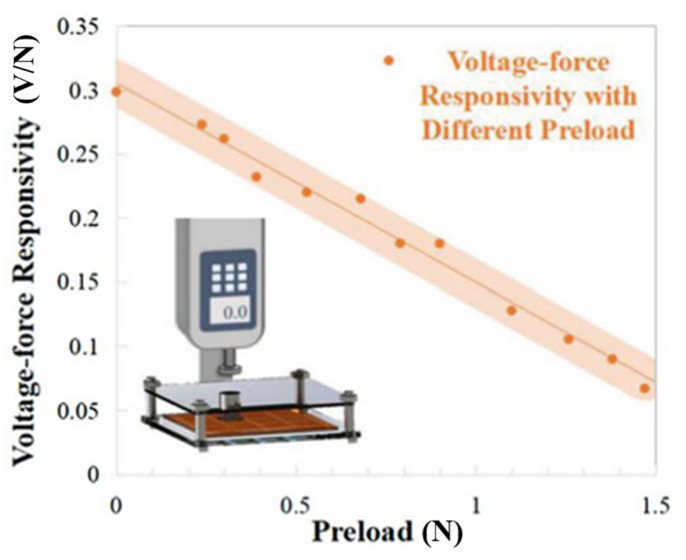
The unstable voltage-force responsivity with different preload [42].

**Figure 11 materials-14-05698-f011:**
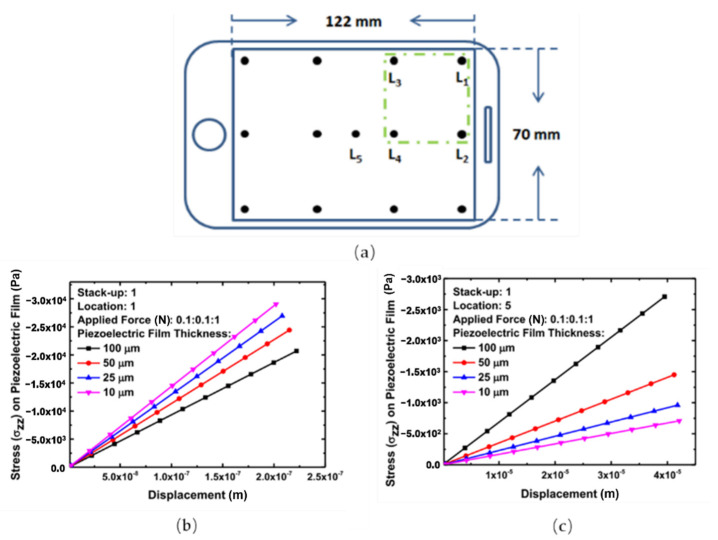
(**a**) The proposed touch panel and investigated touch locations; (**b**) mechanical response of Location 1; (**c**) mechanical response of Location 5 [8].

**Figure 12 materials-14-05698-f012:**
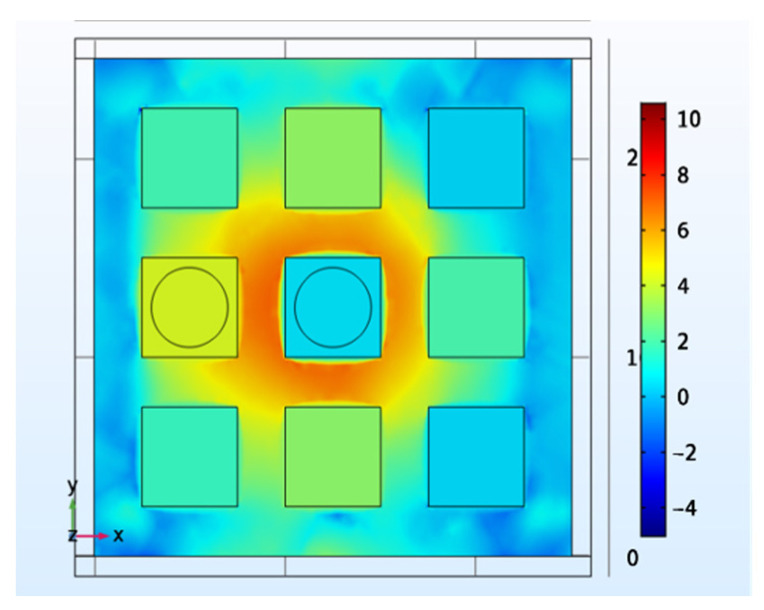
The response when the touch events are applied to adjacent locations simultaneously [9].

**Table 1 materials-14-05698-t001:** Comparison of the commonly used piezoelectric materials [17,18,22,25,35,36,37].

Piezoelectric Materials	Piezoelectric Strain Constant (pC/N)	Piezoelectric Stress Constant (Vm/N)	Relative Permittivity $\varepsilon_{r}$	Electromechanical Coupling Factor (%)	Density (10^3^ kg/m^3^)	Curie Point (°C)	Flexibility	Advantage	Disadvantage
Crystals (Quartz)	$d_{11}$ = 2.3	$g_{11}$ = 0.06	4.65	1	2.65	573	Poor	High $Q_{M}$, low temperature coefficient	Low dielectric constant and piezoelectric coefficient
Crystals (LN)	$d_{33}$ = 6.0	$g_{33}$ = 0.01~0.02	28–85	60	4.65	1140	Poor	Low acoustic losses, colorless, insoluble in water and organic solvents	Phase transition prior to the melting point
Crystals (LT)	$d_{33}$ = 5.7	$g_{33\;}$= 0.01	43–54	30	7.46	605	Poor		
Ceramics (PZT)	$d_{31}$ = 110 $d_{33}$ = 225~590	$g_{31}$ = 0.01 $g_{33}$ = 0.021~0.056	1200	30	7.5	386	Poor	High piezoelectric coefficient	Brittleness, lead contained, high density
Ceramics (BaTiO_3_)	$d_{33}$ = 190	$g_{33}$ = 0.01	1700	35	6	130	Poor	High piezoelectric coefficient and high electromechanical coupling factor, lead-free	Low Curie point
Polymers (PVDF)	$d_{31}=$ 23 $d_{33}=-33$	$g_{31}=$ 0.22 $g_{33}=$ −0.31	12	12	1.78	80	Outstanding	Flexible, light-weight	Low Curie point, small dielectric constant
Ceramic-polymer composite (PZT embedded in an inactive epoxy resin matrix)	$d_{33}$ = 25	$g_{33}$ = 0.09	32	0.5–0.7	4500	-	Good	Flexible, low density compared with ceramics	Low dielectric constant

**Table 2 materials-14-05698-t002:** Typical growth methods for the preparation of BaTiO_3_ thin films [38,39,40,41].

Method	Growth Temperature (°C)	Pressure (Pa)	Substrate	Advantage	Disadvantage
Magnetron sputtering	Room temperature	1	Quartz	Good uniformity; large area	High manufacturing cost
Chemical vapor deposition	600–800	533–667	LaAlO_3_	High purity and density	Requirement for different cavities
Sol-gel method	100 °C for generating the sol; 800 °C for crystallization	Atmospheric pressure	Pt-coated Si wafer	Simple process	Easy to crack
Solvothermal process	140	High pressure	Si (100) deposited with Ti (1000 Å)	Simple operation; low temperature	Danger of high pressure

**Table 3 materials-14-05698-t003:** Drawbacks of the piezoelectric touch panels and the state-of-art solutions.

Drawbacks	Solutions	Performance	Refs
Unstable responsivity introduced by different touch area and direction	Estimating touch area and direction using capacitance information	An improvement in stability of force voltage responsivity of 85%	[46]
	Classifying touch directions using an artificial network	Force detection accuracy of 90%	[45]
$d_{33}$ frequency dependence	Calibrating $d_{33}$ using the frequency information	Improving detection accuracy by 2.5%	[49]
Propagated stress	Eliminating the propagated stress using capacitive information	Eliminating the propagated stress successfully	[50]
Preload effect	Obtaining the prestress using the resonant characteristics and calibrating $d_{33}$	Improving the detection accuracy by 15.17%	[51]
Boundary condition	Estimating touch position and force amplitude using an artificial network	0.56 mm shift of the touch position Force detection accuracy of 94.2%	[52]
	Measuring the curvature radius using the capacitive information and calibrating the response	1.7% change in the force responsivity with 3 cm change in the curvature radius	[53]

**Table 4 materials-14-05698-t004:** Emerging applications based on the piezoelectric touch panels.

Emerging Applications	Principles of the Work	Advantages	Disadvantages	Performance of Typical Design	Refs
Authentication	Extract touch features using piezoelectric touch panel for authentication	Simplicity of use High sensitivity	Decreasing accuracy with the variation of keystroke habit	The EER of 0.720%	[55]
				The accuracy of 97% Multi-user support	[56]
Mood detection	Extract touch behaviors using piezoelectric touch panel for emotion classification	Cost-effective Non-intrusive High sensitivity	Individual difference when experiencing the same emotion	The classification accuracy of 78.31% Short text input	[59]
Underwater	Sensitive to the mechanical displacement Not affected by small amounts of water	No limitations in moist environment	Affected by the temperature of the water	Excellent properties when using in the water	[44]
Humanoid skin	Detect force based on the flexible piezoelectric techniques	Low cost The coupling ability with capacitive techniques	Temperature effect	The force sensitivity of 0.05 N The spatial resolution of 0.29 mm	[65]
Gait analysis	Detect the plantar pressure for gait analysis	Passive detection Low power consumption	The frequency dependency of $d_{33}$	The responsivity of 693.1 mV/N The sensitivity of 0.056 N	[68]

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
