# Peer review of "Piezoelectric Based Touch Sensing for Interactive Displays—A Short Review"

_materials, 2021, doi:10.3390/ma14195698_

Round 1

Reviewer 1 Report

General Remarks:

In my opinion, the Review provides exciting data about the working principles of the piezoelectric touch panel, piezoelectric materials, and their characteristic. Overall, the paper is well written and properly presented. Although the work is interesting, some additions should be carried out and adequately addressed before publication.

Comments:

1) Lines 366-368 should be revised.

2) For the Review to become even more interesting, it is essential to introduce an extra topic showing the most used piezoelectric materials and a brief description of the main techniques used for the growth of these materials. A Table or a diagram would be welcome on this topic.

Author Response

Reviewer: 1

Comments:

In my opinion, the Review provides exciting data about the working principles of the piezoelectric touch panel, piezoelectric materials, and their characteristic. Overall, the paper is well written and properly presented. Although the work is interesting, some additions should be carried out and adequately addressed before publication.

1.Lines 366-368 should be revised.

Reply: Thank you for pointing out this. We have provided more details about the mentioned sensor, including the structure and the extracted parameters.

The content has been revised as follows:

“A flexible sensor based on PVDF is installed in the slipper sole area under the inner arch of the foot. When the subject walks, the normal pressure generates and therefore the piezoelectric sensor under the inner arch of the foot generates output voltage. 8 gait parameters including the time parameters and the number of steps are then extracted from the sensing signals by detecting the heel strike and toe-off times.”

2.For the Review to become even more interesting, it is essential to introduce an extra topic showing the most used piezoelectric materials and a brief description of the main techniques used for the growth of these materials. A Table or a diagram would be welcome on this topic.

Reply: Thank you for this comment. We have supplemented Section “2.3 The Growth of Piezoelectric Materials”. In this section, the commonly used piezoelectric materials and their growth methods are discussed. In addition, we summarize the environment parameters (temperature, pressure, substrate), advantages, and disadvantages of each method in Table 2.

2.3. The Growth of Piezoelectric Materials

“Among the piezoelectric materials, the piezoelectric thin film is the most widely used form, which has the advantages of small volume, lightweight, high working frequency and simple manufacture process of the multi-layer structure. The piezoelectric thin films are widely applied to piezoelectric microelectronic systems and optoelectronic technology fields. The most widely used piezoelectric thin film is of PZT piezoelectric ceramics, which, however, is fragile and highly toxic and can cause big damage to humans. To overcome this problem, lead-free piezoelectric thin films like BaTiO3, KxNa1−xNbO3(KNN), Bi0.5Na0.5TiO3(BNT) are proposed. In addition, piezoelectric thin films based on polymers like PVDF are also receiving great attention due to their good flexibility.

To fabricate piezoelectric thin films, the magnetron sputtering method, sol-gel method, vapor deposition method, and solvothermal method are widely used. [38] Magnetron sputtering refers to the process that bombarding the target with high-speed moving inert particles to deposit atoms on the substrate to form a thin film. This method can form thin films with a large area and good uniformity. However, its high manufacturing cost is not desired. The sol-gel method is to dissolve the metal salt in a common solvent and form a uniform precursor solution (SOL) after hydrolysis and polymerization firstly, then coat the SOL is on a substrate surface. After drying and repeated coating, the film is finally processed by annealing. This method is easier in contrast, but the thin film is easy to crack under high-temperature sintering. The vapor deposition method is to apply the compound or elemental gas to the substrate, and the film is formed by gas-phase action or chemical reaction on the substrate surface. This method can obtain high purity and dense films; however, it is inconvenient because different cavities are required to form different forms. The solvothermal process is to obtain thin films by chemical reaction with organic matter or non-aqueous solvent as the solvent under high temperature and high pressure. It has the advantages of a simple process and low temperature. However, as the process is under high pressure, it is dangerous and needs high-level equipment. The typical growth methods for the preparation of BaTiO3 thin films are shown in table 2.”

Table 2. Typical growth methods for the preparation of BaTiO3 thin films. [38-41]

Method

Growth temperature()

Pressure (Pa)

Substrate

Advantage

Disadvantage

Magnetron sputtering

Room temperature

1

Quartz

Good uniformity; large area

High manufacturing cost

Chemical vapor deposition

600-800

533-667

LaAlO3

High purity and dense

Requirement for different cavities

Sol-gel method

100℃ for generating the sol; 800℃ for crystallization

Atmospheric pressure

Pt-coated Si wafer

Simple process

Easy to crack

Solvothermal process

140

High pressure

Si (100) deposited with Ti (1000 Å)

Simple operation; low temperature

Danger of high pressure;

References:

  1. Li, L.; Miao, L.; Zhang, Z.; Pu, X.; Feng, Q.; Yanagisawa, K.; Fan, Y.; Fan, M.; Wen, P.; Hu, D.J.J.o.M.C.A. Recent progress in piezoelectric thin film fabrication via the solvothermal process. 2019, 7, 16046-16067.
  2. Lee, B.; Zhang, J.J.T.s.f. Preparation, structure evolution and dielectric properties of BaTiO3 thin films and powders by an aqueous sol–gel process. 2001, 388, 107-113.
  3. Maneeshya, L.; Thomas, P.; Joy, K.J.O.M. Effects of site substitutions and concentration on the structural, optical and visible photoluminescence properties of Er doped BaTiO3 thin films prepared by RF magnetron sputtering. 2015, 46, 304-309.
  4. Wills, L.; Wessels, B.W.; Richeson, D.; Marks, T.J.J.A.p.l. Epitaxial growth of BaTiO3 thin films by organometallic chemical vapor deposition. 1992, 60, 41-43.

Reviewer 2 Report

The review article  Piezoelectric Based Touch Sensing for Interactive Displays -A Short Review by Liu et al. is a well presented short review with necessary information provided in a concise package. I would however, want the authors to provide some more information on the new materials that are predicted/or are worked on as future piezoelectric systems for interactive display applications. In addtion, can the authors also provide a short insight into the future of this technology with respect to the carbon footprint and sustainability? After these two minor revisions, I would reccommend the manuscript to be accepted and published. 

Author Response

Reviewer: 2

Comments:

The review article Piezoelectric Based Touch Sensing for Interactive Displays -A Short Review by Liu et al. is a well presented short review with necessary information provided in a concise package. I would however, want the authors to provide some more information on the new materials that are predicted/or are worked on as future piezoelectric systems for interactive display applications. In addtion, can the authors also provide a short insight into the future of this technology with respect to the carbon footprint and sustainability? After these two minor revisions, I would reccommend the manuscript to be accepted and published.

  1. I would however, want the authors to provide some more information on the new materials that are predicted/or are worked on as future piezoelectric systems for interactive display applications.

Reply: Thanks for your kind suggestions. The following contents are added at the end of Section “2.2 Broadly Used Materials” to give more information on the new piezoelectric materials.

“The properties of the commonly used piezoelectric materials mentioned above are shown in Table 1. As for the new piezoelectric materials that can be used in future interactive displays, the flexible piezoelectric materials have become the focus of the research, as one of the important trends of interactive displays is flexible display and wearable devices. In [32], new all-silicone composite materials are reported. Cyanopropyl-modified polysiloxanes and Chloro-modified polysiloxane are crosslinked with non-polar PDMS to produce the materials. Young’s modulus of the highly flexible material is 0.12-0.5MPa, while the average piezoelectric coefficient is 24 pm·V-1, close to that of PVDF. In addition, ceramic-polymer composites are receiving great attention due to their flexibility and good piezoelectric properties. In [33], highly flexible piezoelectric composites with poly dimethyl siloxane (PDMS) using herbal zinc oxide (h-ZnO) as filler are reported. The Young’s modulus of the material is 16 MPa while the piezoelectric coefficient is 29.76 pm/V (h-ZnO 30 wt.%), showing both great flexibility and good piezoelectric properties. In [34], PZT-PDMS composites prepared by solution casting are developed. The piezoelectric coefficient of the material achieves 25pC/N, and Young’s modulus is 4.85 MPa. Hopefully, these highly flexible piezoelectric materials can be applied to the flexible display in the future.”

References:

  1. Racles, C.; Dascalu, M.; Bele, A.; Tiron, V.; Asandulesa, M.; Tugui, C.; Vasiliu, A.-L.; Cazacu, M.J.J.o.M.C.C. All-silicone elastic composites with counter-intuitive piezoelectric response, designed for electromechanical applications. 2017, 5, 6997-7010.
  2. Singh, A.; Das, S.; Bharathkumar, M.; Revanth, D.; Karthik, A.; Sastry, B.S.; Rao, V.R.J.M.R.E. Low cost fabrication of polymer composite (h-ZnO+ PDMS) material for piezoelectric device application. 2016, 3, 075702.
  3. Babu, I.; de With, G.J.C.s.; technology. Highly flexible piezoelectric 0–3 PZT–PDMS composites with high filler content. 2014, 91, 91-97.

  1. Can the authors also provide a short insight into the future of this technology with respect to the carbon footprint and sustainability?

Reply: Thank you for this comment. We have added the following content in Section “7. Conclusions and Perspective” to discuss the future of piezoelectric-based interactive display from the view of sustainable development.

“In addition, as the problems of global warming and environmental pollution are becoming more and more serious, using renewable energy instead of traditional fossil energy to maintain sustainable development is important for the technology. The preferred approach is to use the piezoelectric nanogenerators (PENG) as the power resources to achieve self-powered systems. Recently, Sumera et al reported Br doped 2D ZnO PENG which is used as an active self-powered pressure sensor for the measurement of a wide range of pressure. [76] In the future, it is expected to integrate PENG into the piezoelectric interactive displays to achieve a self-powered system”

References:

  1. Rafique, S.; Kasi, A.K.; Kasi, J.K.; Bokhari, M.; Shakoor, Z.J.C.A.P. Fabrication of Br doped ZnO nanosheets piezoelectric nanogenerator for pressure and position sensing applications. 2021, 21, 72-79.

Reviewer 3 Report

This manuscript reports a short review on the piezoelectric based touch sensing for interactive displays. It presents the work principles of the piezoelectric touch panel, widely-used piezoelectric materials and their characteristic, the applications of the piezoelectric touch panel, as well as the challenges and future trend. I have to say that this manuscript is well written and organized. Therefore, I recommend its publication after addressing the following concerns:

  1. There is a lack of illustrative figures in the "Conventional Applications and Discussions" section. It is highly demanded to insert a figure exemplifying each paragraph to help the reader to understand what the authors are discussing.
  2. In paragraph 2.2, I would recommend to add a paragraph of “ceramic/polymer composites” which is regarded as an interesting category of energy harvesting materials. As mentioned in Table 1, ceramics and polymers process distinct properties, and the composites are regarded to have compromise properties between the ceramics and polymers. See these relevant references:
  • Yao, J. et al. Enhancement of dielectric constant and piezoelectric coefficient of ceramic–polymer composites by interface chelation. Mater. Chem. 19, 2817 (2009).
  • Choi, Y. J. et al. Dielectric and piezoelectric properties of ceramic-polymer composites with 0-3 connectivity type. Electroceramics 30, 30–35 (2013).
  • Hu, D. et al. Strategies to achieve high performance piezoelectric nanogenerators. Nano Energy 55, 288–304 (2019).
  1. In table 1 it is recommended to calculate the g33 coefficient (piezoelectric voltage constant) (see “Choi, Y. J. et al. Dielectric and piezoelectric properties of ceramic-polymer composites with 0-3 connectivity type. Electroceramics 30, 30–35 (2013)” for the calculation).

Author Response

Reviewer: 3

Comments:

This manuscript reports a short review on the piezoelectric based touch sensing for interactive displays. It presents the work principles of the piezoelectric touch panel, widely-used piezoelectric materials and their characteristic, the applications of the piezoelectric touch panel, as well as the challenges and future trend. I have to say that this manuscript is well written and organized. Therefore, I recommend its publication after addressing the following concerns:

1.There is a lack of illustrative figures in the "Conventional Applications and Discussions" section. It is highly demanded to insert a figure exemplifying each paragraph to help the reader to understand what the authors are discussing.

Reply: Thanks for your kind suggestions. We strongly agree that more illustrative figures are needed to make the discussion clearer. The following figures and content are provided in Section “4.1. Conventional piezoelectric-based touch panels”. For saving space, only the new content, figures, and references are provided here. Please refer to the highlighted manuscript for context content.

Figure 8. Three factors leading to unstable force-voltage responsivity. [42]

Figure 9. The propagated stress is related to the detection accuracy. [42]

“In [51], the responsivity with different preload is measured and the results are shown in Figure 10.”

Figure 10. The unstable voltage-force responsivity with different preload. [42]

“In [8], the mechanical response of the different locations is investigated. As shown in Figure 11a, Location 1 is near the edge of the touch panel, while Location 5 is at the center. As the results show in Figure 11b-c, the different boundary conditions result in the different mechanical response, which can reduce the detection accuracy.”

Figure 11. (a) The proposed touch panel and investigated touch locations (b) Mechanical response of Location 1 (c) Mechanical response of Location 5. [8]

References:

  1. Gao, S.; Arcos, V.; Nathan, A. Piezoelectric vs. Capacitive Based Force Sensing in Capacitive Touch Panels. IEEE Access 2016, 4, 3769-3774, doi:10.1109/access.2016.2591535.
  2. Gao, S.; Yan, S.; Zhao, H.; Nathan, A. Touch-Based Human-Machine Interaction: Principles and Applications; Springer Nature: 2021.
  3. Shi, J.; Chen, J.; Lin, J.; Gao, S. Preload Effect Elimination Technique for Piezoelectric Force Touch Sensing in Human-Machine Interactivities. 2020.

  1. In paragraph 2.2, I would recommend to add a paragraph of “ceramic/polymer composites” which is regarded as an interesting category of energy harvesting materials. As mentioned in Table 1, ceramics and polymers process distinct properties, and the composites are regarded to have compromise properties between the ceramics and polymers. See these relevant references:

Yao, J. et al. Enhancement of dielectric constant and piezoelectric coefficient of ceramic–polymer composites by interface chelation. Mater. Chem. 19, 2817 (2009).

Choi, Y. J. et al. Dielectric and piezoelectric properties of ceramic-polymer composites with 0-3 connectivity type. Electroceramics 30, 30–35 (2013).

Hu, D. et al. Strategies to achieve high performance piezoelectric nanogenerators. Nano Energy 55, 288–304 (2019).

Reply: Thank you for your kind suggestions and for bringing us these representative works. The following content and references have been added in Section “2.2.4. Ceramic-polymer Composite”. In addition, Table 1 has been updated with the information on piezoelectric-ceramic composite.

2.2.4. Ceramic-polymer Composite

“As discussed before, the piezoelectric ceramics process high piezoelectric coefficients; however, the poor flexibility of ceramics limits their application. In comparison, the piezoelectric polymers are much more flexible but have a smaller piezoelectric coefficient. To meet the requirement of both good flexibility and high performance, researchers have proposed ceramic-polymer composites, in which the ceramic phase is dispersed in a polymer matrix. [25] This kind of material shows excellent properties of good flexibility, high piezoelectric coefficient, as well as low density. Moreover, in [26], an interfacial chelation mechanism between PZT and a chelating polymer is reported, which can improve the dielectric constant and piezoelectric coefficient of ceramic-polymer composites significantly.

Ceramic-polymer composites are widely used in many fields. Considering its flexibility and good piezoelectric properties, they are applied to piezoelectric nanogenerators with high output performance. [27] In addition, as the polymer phase lowers the density and dielectric constant, ceramic-polymer composites are easier to realize acoustic impedance matching compared to ceramics, and therefore they are widely used in hydrophone, medical examination probes, and sonar.” [28]

Round 2

Reviewer 3 Report

All the changes requested by the reviewer have been introduced by the authors. Therefore, I propose the acceptance of the manuscript in its current form.